# A *Vibrio cholerae* viral satellite maximizes its spread and inhibits phage by remodeling hijacked phage coat proteins into small capsids

Caroline M Boyd[1], Sundharraman Subramanian[2], Drew T Dunham[1], Kristin N Parent[2], Kimberley D Seed[1]*

[1]Department of Plant and Microbial Biology, Seed Lab, University of California, Berkeley, Berkeley, United States; [2]Department of Biochemistry and Molecular Biology, Parent Lab, Michigan State University, East Lansing, United States

*For correspondence:
kseed@berkeley.edu

**Abstract** Phage satellites commonly remodel capsids they hijack from the phages they parasitize, but only a few mechanisms regulating the change in capsid size have been reported. Here, we investigated how a satellite from *Vibrio cholerae*, phage-inducible chromosomal island-like element (PLE), remodels the capsid it has been predicted to steal from the phage ICP1 (Netter et al., 2021). We identified that a PLE-encoded protein, TcaP, is both necessary and sufficient to form small capsids during ICP1 infection. Interestingly, we found that PLE is dependent on small capsids for efficient transduction of its genome, making it the first satellite to have this requirement. ICP1 isolates that escaped TcaP-mediated remodeling acquired substitutions in the coat protein, suggesting an interaction between these two proteins. With a procapsid-like particle (PLP) assembly platform in *Escherichia coli*, we demonstrated that TcaP is a bona fide scaffold that regulates the assembly of small capsids. Further, we studied the structure of PLE PLPs using cryogenic electron microscopy and found that TcaP is an external scaffold that is functionally and somewhat structurally similar to the external scaffold, Sid, encoded by the unrelated satellite P4 (Kizziah et al., 2020). Finally, we showed that TcaP is largely conserved across PLEs. Together, these data support a model in which TcaP directs the assembly of small capsids comprised of ICP1 coat proteins, which inhibits the complete packaging of the ICP1 genome and permits more efficient packaging of replicated PLE genomes.

## eLife assessment

This **valuable** study reports on the structure and function of capsid size-determining external scaffolding protein encoded by a *Vibrio* phage satellite. The structural work is of high quality and the presented reconstructions are **compelling**. The paper offers a substantial advance in the field of phage and virus structure and assembly, with implications for understanding the evolution of phage satellites.

## Introduction

Like the parasitism of bacteria by the phages that infect them, phages are parasitized by mobile genetic elements termed phage satellites. Satellites continue to be discovered across bacterial species and, despite the independent evolution of these elements, common themes have emerged (reviewed in *de Sousa et al., 2022*). Universally, satellites are integrated genetic elements that are dependent on infection by a helper phage for their lifecycle. Helper phage infection triggers satellites' genome

excision and replication. Following genome replication, satellites parasitize the helper phages' structural components and selectively package satellite genomes into proteinaceous shells comprised of coat proteins, called capsids (*Christie and Dokland, 2012*). The piracy of essential phage structural proteins affords the phage satellites the luxury of reducing their coding capacity and, therefore, their genome size. As such, satellites' genomes can be packaged into smaller capsids compared to their helper phages. In all documented cases, the satellite encodes a strategy to direct the assembly of small capsids, a mechanism that excludes the complete packaging of the larger helper phage genome but permits the complete packaging of the smaller phage satellite genome (*Shore et al., 1978*; *Damle et al., 2012*; *Hawkins et al., 2021*; *Alqurainy et al., 2022*). After the attachment of tails, also pirated from the helper phage, the mature virions harboring the satellite genome are released from the cell. Here, they transduce and integrate the satellite genome into neighboring susceptible bacteria.

Phage-inducible chromosomal island-like elements (PLEs) are satellites of the lytic *Vibrio cholerae* phage ICP1 (*O'Hara et al., 2017*). PLEs are one of four families of phage satellites that have been identified to date (*de Sousa et al., 2022*). In addition to their specificity to *V. cholerae* and unique complete inhibition of their helper phage, PLEs stand apart from other satellites in their genetic composition. They encode genes with similar functions, but without sequence identity, to those in other satellites. To date, 10 genetically distinct but related PLEs have been identified (*Angermeyer et al., 2021*), with PLE1 being the most well studied. PLEs, like other satellites, have a smaller genome (~18 kb) than the phage they parasitize, ICP1 (~125 kb), and are dependent on their helper for excision (*McKitterick and Seed, 2018*; *Nguyen et al., 2022*), replication (*Barth et al., 2020*), and virion production (*Netter et al., 2021*). PLE1 has been shown to produce virions with small ~50-nm-wide capsids (*Figure 1A*), while ICP1 produces virions with large ~80-nm-wide capsids (*Figure 1B*). Based on similarities to other capsid-remodeling satellites and the evidence that depletion of ICP1's coat protein by CRISPRi during infection results in reduced PLE transduction (*Netter et al., 2021*), it is hypothesized that PLE capsids are comprised of ICP1 coat proteins. However, the mechanism that PLE1 uses to achieve this capsid remodeling has yet to be explored.

As capsids for large double-stranded DNA viruses assemble through a stepwise pathway, remodeling must occur at the first stage, nucleation of the procapsid. Procapsids are the empty shell capsid precursors into which DNA is packaged (*Steven et al., 2005*). Procapsid size, referred to by a T number (which corresponds to the number of triangular structural units within a face of an icosahedron and largely represents the size of the capsid), and assembly are regulated by scaffolding proteins that guide coat proteins into their correct orientation around the pre-formed portal complex (*King and Casjens, 1974*; *Dokland, 1999*). These capsid scaffolds can either be separately encoded proteins or contained within a domain of the coat protein, as is seen in the phage HK97 (*Prevelige et al., 1988*; *Duda et al., 1995*). So, to alter the size of the capsids, satellites must regulate the size of the procapsid.

Highlighting the convergent evolution across unique satellite families, four distinct strategies to produce small capsids have been characterized. Two representatives from different families of satellites exploit the role of scaffolds in the capsid assembly process to promote the formation of smaller capsids. First, some of the *Staphylococcus aureus* pathogenicity islands (SaPIs) encode an alternative internal scaffold, CpmB, that binds to the helper phage's coat proteins inside of the assembling procapsid (*Dearborn et al., 2017*). CpmB increases procapsid curvature, resulting in the smaller procapsid (*Dearborn et al., 2017*). The second example of capsid remodeling was described in the *Escherichia coli* phage satellite, P4 (*Shore et al., 1978*). Analogous to the satellite-encoded scaffold from SaPIs, P4 encodes an alternative scaffold, Sid, that functions to regulate assembly of the smaller procapsid; however, Sid binds to the outside of the procapsid (*Kizziah et al., 2020*; *Marvik et al., 1995*). Sid proteins form an external cage around the helper phage's coat proteins which promotes the assembly of the small procapsids (*Kizziah et al., 2020*). An alternative strategy to redirect capsid size is found in a subfamily of SaPIs, including SaPIbov5, which encodes a homolog of their helper phage's coat protein that exclusively forms the pentamers while the helper phage's coat proteins form the hexamers of the small capsid (*Hawkins et al., 2021*). These coat proteins encode their scaffolds within a domain of the coat, like HK97 (*Hawkins et al., 2021*). Exactly how the satellite-derived coat pentamers promote the assembly of small icosahedral capsids instead of the larger prolate capsid of the helper is not fully understood. The last characterized strategy used by satellites to make small capsids was recently discovered in the aptly named capsid-forming phage-inducible chromosomal

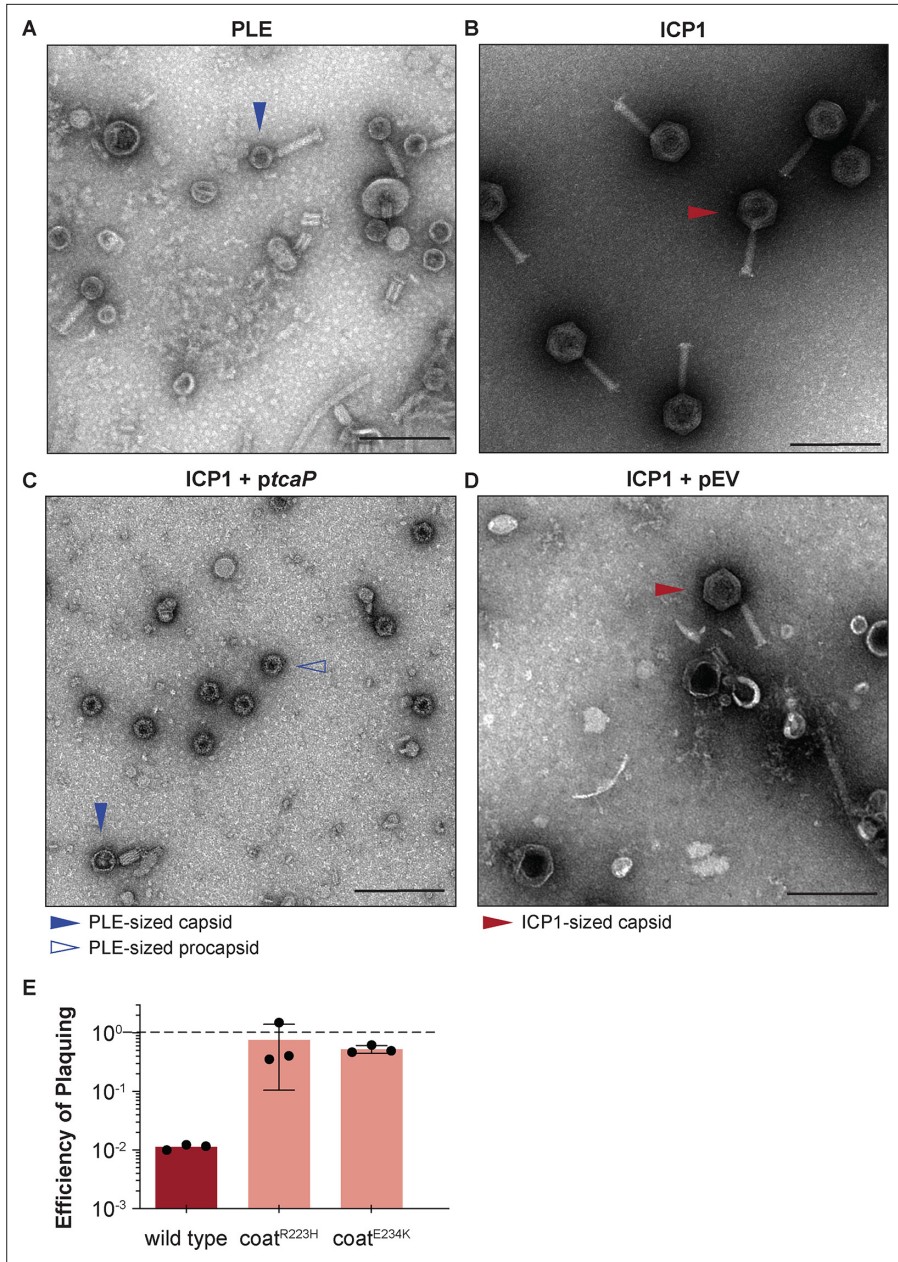

**Figure 1.** Phage-inducible chromosomal island-like element (PLE) encodes an inhibitory protein, TcaP, which modifies ICP1's capsid assembly process to produce small capsids. (**A, B**) Representative transmission electron micrographs (TEMs) from three independent biological replicates show (**A**) PLE virions have small, ~50-nm-wide capsids and long contractile tails while. (**B**) ICP1 virions have large, ~80-nm-wide capsids and long contractile tails. Scale bars are 200 nm. (**C, D**) Representative TEMs of lysates produced from ICP1 infection of *V. cholerae* expressing (**C**) TcaP (p*tcaP*) or (**D**) an empty vector (pEV). Arrowheads show exemplary capsids and their sizes according to the legend. Scale bars are 200 nm. (**E**) Efficiency of plaquing of wild type ICP1 or escape phages harboring the substitution indicated in the coat protein on *V. cholerae* expressing TcaP relative to an empty vector control. Each dot represents a biological replicate, bars represent the mean, and error bars show the standard deviation. The dotted line indicates an efficiency of plaquing of 1 where the expression of TcaP is not inhibitory to plaque formation.

The online version of this article includes the following source data and figure supplement(s) for figure 1:

**Source data 1.** Data used to create *Figure 1E*.

**Source data 2.** Uncropped micrographs used to create *Figure 1A–D*.

*Figure 1 continued on next page*

*Figure 1 continued*

**Figure supplement 1.** More than one particle morphology is seen in lysates produced from ICP1 infection of *V. cholerae* with an empty vector or expressing *tcaP*.

**Figure supplement 1—source data 1.** Uncropped micrographs used to create *Figure 1—figure supplement 1A and B*.

islands (cfPICIs), which avoid altering the assembly of their helper phage capsids by instead encoding their own structural components of the capsid and stealing only the phage tails to construct satellite virions (*Alqurainy et al., 2022*). Interestingly, despite the divergence in capsid remodeling strategies, there is convergence on capsid size where each satellite assembles icosahedral capsids with 240 subunits, or T = 4 capsids. PLE's small capsids are similar in size to these T = 4 capsids but PLE does not encode obvious homologs of CpmB or Sid nor capsid homologs, making it unclear how capsid remodeling is achieved in this divergent satellite. Notably, other helper phages have similar genome sizes, ~30–45 kb, while ICP1 has a significantly larger genome, ~125 kb and is the only known helper packaged into T = 13 capsids (*Boyd et al., 2021*; *Depelteau et al., 2022*). This may suggest that PLE uses a unique mechanism to remodel the ICP1 coat proteins into the smaller ~50-nm-wide PLE-sized capsids made from a considerably smaller number of subunits.

Here, we studied how PLE remodels the ICP1 capsid. We found a PLE-encoded protein that has an essential role in making small capsids, named TcaP for its *tiny capsid* phenotype. TcaP's activity was necessary and sufficient to make small capsids, which inhibited the production of infectious ICP1 virions and increased the efficiency of PLE transduction. We studied TcaP's mechanism of capsid remodeling using a heterologous procapsid-like particle (PLP) assembly platform in *E. coli* and cryo-genic electron microscopy (cryo-EM). These data revealed TcaP as an external scaffolding protein that directs the assembly of ICP1 coat proteins into small, PLE-sized capsids. Finally, we analyzed the known PLE variants and found TcaP is largely conserved in PLEs. Our work uncovered the mechanism of capsid remodeling in the phage satellite PLE and provides the first example of small capsid production benefiting satellite transduction.

## Results

### PLE encodes a single gene product, TcaP, that is necessary and sufficient to direct the assembly of small capsids

Given the observation that PLE virions have smaller capsids than ICP1 virions (*Figure 1A and B*), we hypothesized that PLE encodes a single protein responsible for assembling these smaller capsids. The generation of small capsids would limit the amount of DNA that could be accommodated within the capsids and block the complete packaging of the larger ICP1 genome, reducing ICP1 plaquing. In line with this prediction, other satellites' proteins that redirect capsid assembly are inhibitory to their helper phages for this reason (*Shore et al., 1978*; *Damle et al., 2012*; *Carpena et al., 2016*). Using this logic and PLE1 as the representative PLE, we identified one candidate PLE protein, TcaP (previously Orf17 (AGG09411.1)), as a putative capsid remodeling protein.

First, we used transmission electron microscopy (TEM) to evaluate the morphology of virions produced after a single round of ICP1 infection in a PLE(-) *V. cholerae* strain expressing *tcaP* from a plasmid (p*tcaP*). Lysates collected from the p*tcaP* host showed an abundance of small, ~50-nm-round particles reminiscent of PLE-sized procapsids (*Figure 1C*). Rarer particles in these samples resembled virions produced from a PLE(+) infection with small capsids and attached tails, though apparently lacking DNA (*Figure 1—figure supplement 1A*). In comparison, many virions produced from the empty vector control were ICP1-sized virions with large capsids, as expected (*Figure 1D*). Other particles observed in the control samples were full ICP1 capsids without tails, empty ICP1 procapsids, or empty expanded capsids (*Figure 1—figure supplement 1B*). Importantly, these data demonstrate that *tcaP* expression during ICP1 infection, in the absence of other PLE-encoded products, results in the production of small capsids.

In agreement with the expectation that small capsid formation would be inhibitory to ICP1, p*tcaP* inhibited ICP1 plaque formation by 100-fold (*Figure 1E*). Rare plaques isolated on the p*tcaP* strain were picked and serially propagated on the restrictive strain. Two escape phages were whole-genome

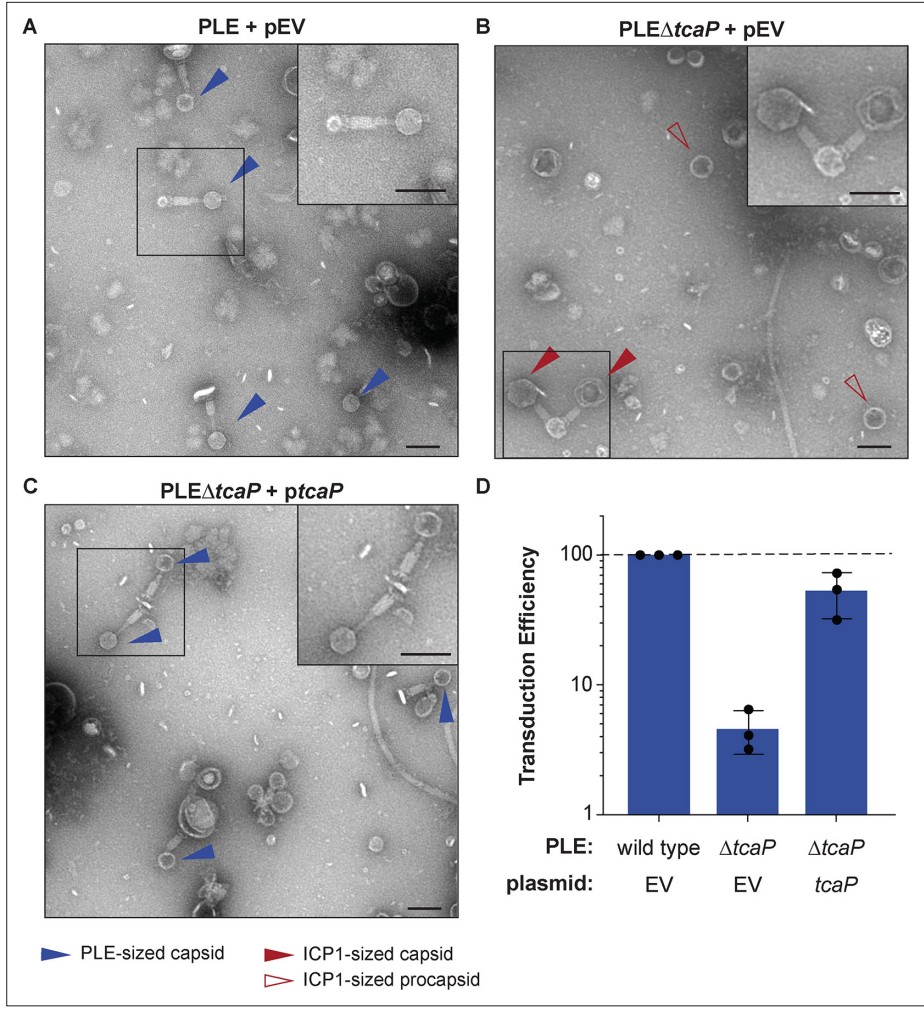

**Figure 2.** TcaP is the only phage-inducible chromosomal island-like element (PLE)-encoded factor necessary for directing small capsid assembly, which is required for efficient PLE transduction. (**A–C**) Representative transmission electron micrographs (TEMs) from three independent biological replicates of lysate from ICP1-infected strains of *V. cholerae* with wild type PLE or PLEΔ*tcaP*, as indicated, carrying either the empty vector (EV) or a vector expressing TcaP (*tcaP*). Insets are enlarged regions of the images highlighting representative particles, and arrowheads indicate capsid types and sizes, as described in the legend. Scale bars are 100 nm. (**D**) Quantification of PLE genome transduction for the strain indicated represented as the transduction efficiency relative to wild type PLE with an empty vector (pEV). Each dot represents a biological replicate, bars represent the mean, and error bars show standard deviation. The dotted line indicates an efficiency of 100%.

The online version of this article includes the following source data for figure 2:

**Source data 1.** Data used to create *Figure 2D*.

**Source data 2.** Uncropped micrographs used to create *Figure 1A–C*.

sequenced, and each harbored a single nonsynonymous mutation in the gene encoding the coat protein, resulting in substitutions R223H or E234K (*Supplementary file 1*). Substitutions at unique but proximal positions in coat protein from independent escape phages suggest that TcaP interacts with the coat protein, an expected feature of TcaP's predicted role as a capsid remodeling protein. Collectively, these genetic escapes, coupled with TcaP's inhibition of ICP1 plaquing, which is accompanied by the production of small capsids, strongly support that TcaP is a capsid remodeling protein.

Expression of TcaP during ICP1 infection was sufficient to redirect capsid size, but other capsid remodeling satellites have been shown to use either a single protein (*Shore et al., 1978*; *Carpena et al., 2016*) or two proteins that act in concert to direct small capsid assembly (*Damle et al., 2012*). Virions resulting from ICP1 infection of wild type PLE and PLEΔ*tcaP* were concentrated by

centrifugation and their morphology was assessed by TEM. The wild type PLE control produced PLE-sized virions with small capsids as expected and no ICP1-sized virions were observed (*Figure 2A*). The PLEΔ*tcaP* strain produced only virions with large, ICP1-sized capsids and no PLE-sized particles (*Figure 2B*), a phenotype that was rescued following the expression of *tcaP in trans* (*Figure 2C*). Importantly, *tcaP* is dispensable for PLE-mediated inhibition of ICP1 (*Hays and Seed, 2020*), indicating that the increase in capsid size in the absence of TcaP is specific to its role in capsid-size redirection and not a result of ICP1 escaping PLE. These data demonstrate that TcaP is necessary for PLE's redirection of ICP1 coat protein into small capsids and that PLE does not encode a redundant mechanism for capsid remodeling.

Next, we measured the transduction of PLE's genome following ICP1 infection in PLEΔ*tcaP* hosts using a previously described assay using an antibiotic resistance marker in PLE (*O'Hara et al., 2017*). Interestingly, in the absence of *tcaP*, PLE transduction was tenfold less efficient than wild type (*Figure 2D*). The defect in the transduction of the PLEΔ*tcaP* strain was largely restored with p*tcaP* (*Figure 2D*). Together with the morphological data of transducing particles, these data suggest that TcaP-mediated capsid remodeling facilitates more efficient horizontal spread of the PLE genome to recipient *V. cholerae*. PLE is the first satellite to show dependency on small capsids for efficient transduction.

## PLE-encoded TcaP is a bona fide capsid scaffold

Capsid scaffolding proteins are characteristically responsible for directly promoting the assembly of coat proteins and controlling capsid size (*Dokland, 1999*). TcaP expression results in the formation of small capsids during ICP1 infection (*Figure 1C*), which suggests that it has scaffolding activity. However, it is possible that TcaP does not act directly on the coat proteins as a scaffold, but rather interferes with ICP1's capsid morphogenesis pathway in some way that decreases capsid size. To directly address if TcaP is a scaffold, we set up a heterologous PLP assembly platform in *E. coli* similar to those previously described (*Spilman et al., 2012*; *Wang et al., 2006*; *Cerritelli and Studier, 1996*). Briefly, we co-expressed ICP1's coat and putative scaffolds from either ICP1 or PLE and monitored the production of PLPs. By adding a C-terminal six-histidine (6xHis) tag to the coat protein, we were able to purify coat-containing complexes by affinity chromatography, examine their protein content by SDS-PAGE/Coomassie staining, and their morphology by TEM. First, as a control, coat::6xHis (referred to as 'coat' for simplicity) was expressed and purified. The coat proteins eluted as complexes, but they were irregular in size and shape, often forming spirals (*Figure 3A1*, *Figure 3—figure supplement 1*), as is expected for coat proteins in the absence of scaffolds (*Dokland, 1999*). As a second control, we confirmed that ICP1's putative scaffold could assemble coat proteins into uniform, ICP1-sized PLPs (*Figure 3A2*, *Figure 3—figure supplement 1*). The presence of scaffold in these PLPs was supported by SDS-PAGE analysis in which we observed a band corresponding to the predicted size of the scaffold (39.3 kDa), indicating, as expected, that the scaffold co-eluted with coat-containing complexes (*Figure 3A2*). Interestingly, two smaller bands of approximately 23 and 17 kDa also appeared in these samples. The size of these bands is consistent with the cleavage of the scaffold resulting in two fragments. As scaffolds are generally not a part of the mature capsid, they are removed from the procapsid by self-cleavage (*Robertson et al., 1997*; *Chang et al., 2008*) or cleavage by a protease (*Duda et al., 2013*; *Huet et al., 2016*). Indeed, in the presence of protease inhibitors during purification, these bands were not observed, but the same-sized particles were produced, suggesting that scaffold cleavage is not a prerequisite for assembly (*Figure 3—figure supplement 2A*). Analysis of ICP1 PLPs by mass spectrometry (LC-MS/MS) further supported the presence of ICP1's putative scaffold (*Supplementary file 2*). Together, these data confirm the predicted role of ICP1's scaffold and demonstrate that PLPs can be assembled and purified using this platform.

Other known satellite systems have varying dependencies on the phage-encoded scaffold and/or additional satellite-encoded factors for the assembly of small procapsids (*Damle et al., 2012*; *Chang et al., 2008*; *Marvik et al., 1994*; *Six, 1975*). To test whether TcaP requires ICP1's scaffold to produce small PLPs, we co-expressed coat with either just TcaP or with both TcaP and ICP1's scaffold simultaneously. First, analysis of the stained SDS-PAGE gel following co-expression of coat and TcaP showed that TcaP bound to coat and was co-eluted (*Figure 3A3*). TEMs showed that TcaP was sufficient to assemble ICP1's coat into PLPs that were homogenously PLE-sized (*Figure 3A3*, *Figure 3—figure supplement 1*). These data demonstrate that TcaP has scaffolding activity for ICP1's coat protein and

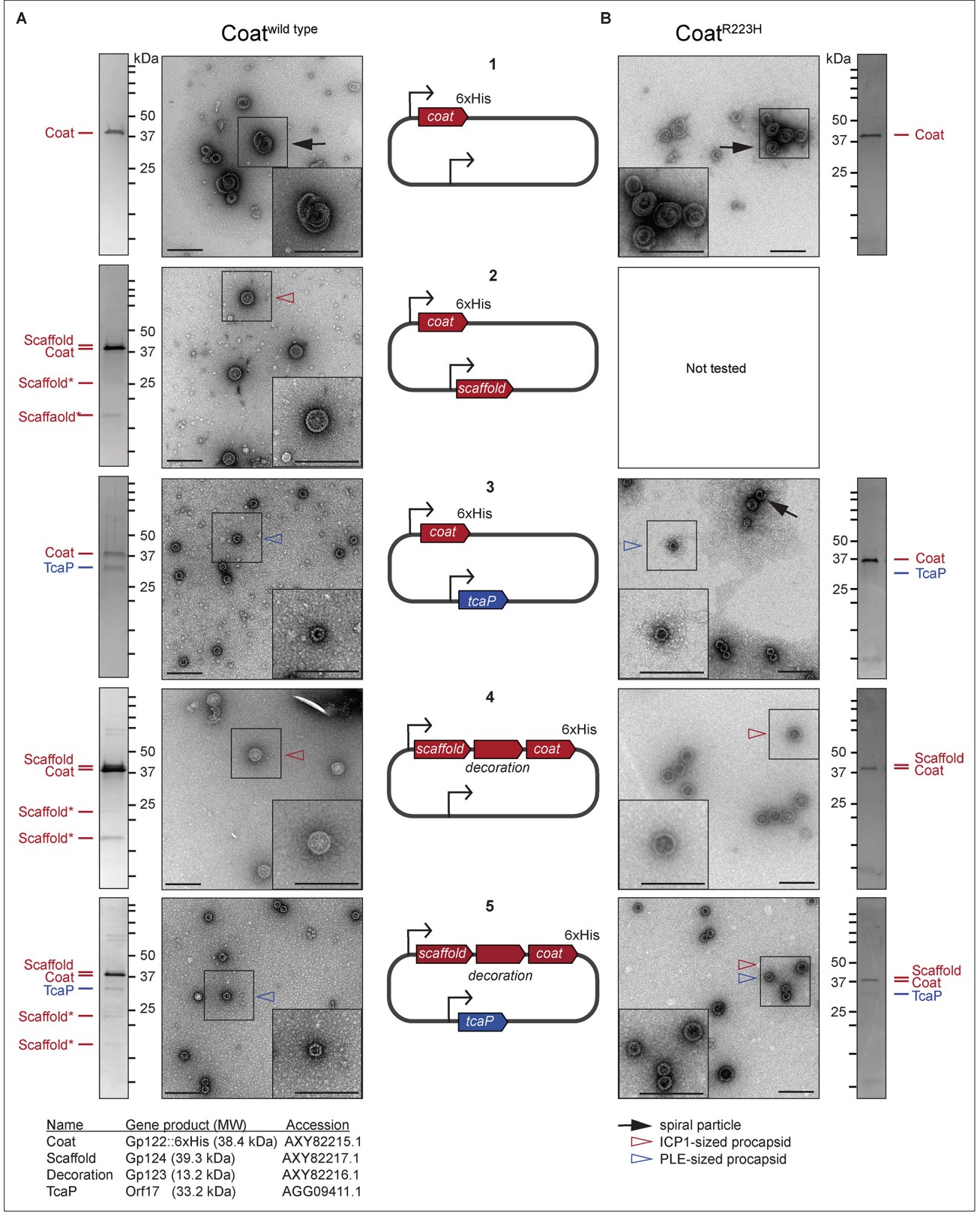

**Figure 3.** The size of procapsid-like particles (PLPs) is determined by ICP1 and phage-inducible chromosomal island-like element (PLE) scaffolds. Representative transmission electron micrographs (TEMs) and Coomassie stained SDS-PAGE analyses of resulting affinity-purified PLPs produced in the heterologous assembly platform in *E. coli* from 2 to 3 independent biological replicates following expression of (**A**) coat^wild type or (**B**) coat^R223H proteins encoded on plasmids as shown in the diagrams in the central panel (numbered 1–5). ICP1-encoded genes are shown in red and PLE-encoded genes are

*Figure 3 continued on next page*

*Figure 3 continued*

shown in blue. Bent arrow icons indicate P$_{tac}$ promoters. Six-histidine (6xHis) represents the tag fused to the C-terminus of the coat. Protein standards are indicated by black tick marks and a subset are marked by their sizes in kDa as indicated (standard range: 250, 150, 100, 75, 50, 37, 35, 20, 15, 10 kDa). Protein bands of interest are indicated by colored tick marks and labels (see legend for calculated molecular weights and accession numbers of these proteins, complete gene and protein information is provided in the Key resources table). In the absence of protease inhibitors, ICP1's scaffold appears to be cleaved and the resulting cleavage products are indicated by scaffold*. Protease inhibitors were included in all coat$^{R223H}$ purifications. TEM insets are enlarged sections of the images highlighting representative particles and arrowheads indicate capsid types and sizes, as described in the legend. Scale bars are 200 nm.

The online version of this article includes the following source data and figure supplement(s) for figure 3:

**Figure supplement 1.** Quantification of size and shape of procapsid-like particle produced in *E. coli*.

**Figure supplement 1—source data 1.** Data used to create *Figure 3—figure supplement 1A and B*.

**Figure supplement 2.** ICP1's scaffold is cleaved in the absence of protease inhibitors and procapsid-like particle (PLP) assembly is not dependent on scaffold cleavage.

**Figure supplement 2—source data 1.** Uncropped micrographs and gels used to create *Figure 3—figure supplement 2A and B*.

**Figure supplement 3.** ICP1's decoration protein is produced but not assembled into procapsid-like particles (PLPs).

**Figure supplement 3—source data 1.** Uncropped micrographs and gels used to create *Figure 3—figure supplement 3*.

**Figure supplement 4.** The E234K substitution diminishes assembly of the coat protein in the procapsid-like particle (PLP) assembly platform.

**Figure supplement 4—source data 1.** Micrographs and gels used to create *Figure 3—figure supplement 4*.

that it does not require additional factors to assemble small PLPs. Next, we constructed a plasmid expressing the scaffold and the coat as they occur natively in ICP1's genome downstream of one of the promoters. Here, the gene *gp123* is present between the scaffold and coat and predicted to encode for the decoration protein. The putative decoration protein was not expected to be incorporated into the PLPs as decoration proteins are typically added after genome packaging, and capsid expansion exposes their binding site, as has been shown for phage Lambda (*Lander et al., 2012*). In line with this, the stained gel of the resulting purified ICP1-sized PLPs indicated that while the ICP1 scaffold co-eluted with coat-containing complexes as expected, the decoration protein was produced but not incorporated into the assembled particles (*Figure 3A4*, *Figure 3—figure supplement 1*). Interestingly, when we co-expressed ICP1 coat, decoration, and scaffold along with TcaP, we did not observe any large particles, suggesting that TcaP is dominant over ICP1's scaffold (*Figure 3A5*, *Figure 3—figure supplement 1*). SDS-PAGE analyses of particles produced in the presence of ICP1's scaffold and TcaP confirmed the presence of TcaP as well as the phage-encoded scaffold and its cleavage products, suggesting that TcaP does not block the incorporation of ICP1's scaffold into procapsids (*Figure 3A5*). This experiment was repeated in the presence of protease inhibitors which eliminated the ICP1 scaffold cleavage products but reproduced the small particle morphology (*Figure 3—figure supplement 2B*). Collectively, these data demonstrate scaffolding activity for both ICP1's scaffold and PLE's TcaP and show that TcaP can assemble small PLPs in the absence or presence of ICP1's scaffold.

Having observed that TcaP is sufficient to make small PLPs in the heterologous assembly platform as well as when expressed from a plasmid during ICP1 infection, we next tested coat$^{R223H}$ and coat$^{E234K}$, the previously identified in vivo genetic escapes of TcaP activity (*Figure 1E*), in the assembly platform. The coat$^{E234K}$ variant was not assembly competent in our heterologous assembly platform (*Figure 3—figure supplement 4*), so we only continued with the coat$^{R223H}$ variant. As expected, in the absence of any scaffolding proteins, coat$^{R223H}$ formed similar spiral complexes as those observed with the wild type coat (*Figure 3B1*, *Figure 3—figure supplement 1*). These data demonstrate that the R223H substitution does not compromise the protein's ability to bind to itself, nor does the substitution provide a means for the coat to form PLPs independent of a scaffold. Next, we expressed coat$^{R223H}$ with TcaP, anticipating we would observe the formation of large ICP1-sized PLPs because this substitution was sufficient for ICP1 to avoid TcaP's inhibitory activity (*Figure 1E*). Unexpectedly, TcaP still directed the assembly of small PLPs comprised of coat$^{R223H}$ (*Figure 3B3*, *Figure 3—figure supplement 1*). However, TcaP was not robustly evident in the purified complexes as assessed by SDS-PAGE, and some particles were spirals, similar to those seen when the coat is expressed without a scaffold (*Figure 3B3*, *Figure 3—figure supplement 1*). These data suggest that TcaP's scaffolding activity was partially compromised when the coat protein carried the R223H substitution. We hypothesized that with ICP1's scaffold the coat$^{R223H}$ proteins could be assembled into ICP1-sized PLPs. Leveraging

the dual expression from a single promoter (*Figure 3A4*), we confirmed that the ICP1 scaffold alone directed the assembly of ICP1-sized PLPs with coat[R223H] (*Figure 3B4*, *Figure 3—figure supplement 1*), but we observed a mix of PLE and ICP1-sized PLPs following addition of TcaP (*Figure 3B5*, *Figure 3—figure supplement 1*). We can attribute this phenotype to the substitution in the coat and not to the presence of protease inhibitors used during the purification as the addition of protease inhibitors during purification of the coat[wild type] co-expressed with the ICP1 scaffold and TcaP resulted in only small PLPs (*Figure 3—figure supplement 2B*). These data suggest that the R223H substitution in ICP1's coat can only partially escape TcaP-mediated small capsid formation, but in vivo this level of escape is sufficient to allow for approximately equal levels of viable progeny production as is seen in the absence of TcaP.

## Cryo-EM reveals TcaP is an external scaffold

The data obtained from the assembly of PLPs in *E. coli* demonstrate that TcaP has scaffolding activity; however, these data do not distinguish between TcaP acting as an internal or an external scaffold. Therefore, we validated the protein content of PLE PLPs produced from co-expression of TcaP and coat by LC-MS/MS and subjected them to cryo-EM. As anticipated, LC-MS/MS confirmed PLE PLPs were comprised of coat and TcaP (*Supplementary file 2*). Representative micrographs from negatively stained samples (*Figure 4A*) and vitrified particles (*Figure 4B*) show a distinctly bumpy surface on the PLE PLPs, suggestive of external proteins. A total of 379,643 particles were used for icosahedral reconstruction of the PLE PLP particle, with a final resolution of 3.4 Å (*Figure 4C*, *Figure 4—figure supplement 1*). The external density of the PLPs corresponded to TcaP, demonstrating that it functions as an external scaffold. Internal density was assigned to coat proteins with a canonical HK97-like fold, which exist as pentamers and hexamers within the 48 nm, T = 4 icosahedral PLP. We used AlphaFold2 (*Jumper et al., 2021*) to predict the structures of the coat protein and TcaP, which were fitted into density from the cryo-EM and aided in modeling nearly all of the coat protein and approximately half of TcaP (residues 34–172) for which only a few side chains could be modeled. As expected, the A-domains of the coat proteins were centered in both pentamers and hexamers. TcaP dimers meet and form trimeric interactions at the threefold axes between hexamers. Broadly, the reconstruction of PLE PLPs shows TcaP functions as an external scaffold and provides context for how TcaP's higher-order structure regulates the number of coat proteins that can be accommodated within the assembling PLP.

Further, the reconstruction reveals details of the residues that were substituted in phages that escape TcaP-mediated capsid remodeling (R223H and E234K). Both residues are found in the A-domain which is oriented near the center of the hexamer where TcaP binds (*Figure 4D*). The arginines at position 223 were clearly visible in the map. Due to the twofold symmetry in the hexamer, there are identical interactions between each half of the hexamer and the TcaP dimer. Importantly, there are three distinct interactions between TcaP and R223 (*Figure 4E*). First, R223 from coat_D is in proximity to TcaP's residues 87–93, while coat_B has R223 positioned near TcaP's residues 63–70, and R223 from coat_C lies close to the second TcaP subunit. In this third interaction, the negatively charged aspartic acid at position 218 from the coat coordinates with the positive charges from the arginine residues from coat (R223) and TcaP (R133) and creates a salt bridge with residues measuring less than 4 Å apart. The electrostatics of the complex, with the TcaP dimer oriented as in *Figure 4D and F*, show TcaP's R133 fitting into a negatively charged pocket on the coat (*Figure 4G*). The side chain for the other residue substituted in the escape phage, E234, could not be confidently modeled, but its localization within the capsid is outside of the TcaP binding region. The exact nature of E234 is not yet clear, but perhaps it has a role in stabilizing the A-domain. Together, these data support the conclusion that ICP1 escapes TcaP's scaffolding activity by altering the coat to affect TcaP's binding site.

If ICP1 is under selective pressure to escape PLE-mediated capsid remodeling in nature, ICP1 isolates could be expected to harbor these coat substitutions. However, comparison of the coat alleles from 67 genetically distinct ICP1 isolates revealed no differences within the A-domain of the alleles. These data show that ICP1 does not naturally encode mutations in the A-domain that would escape TcaP-mediated capsid remodeling.

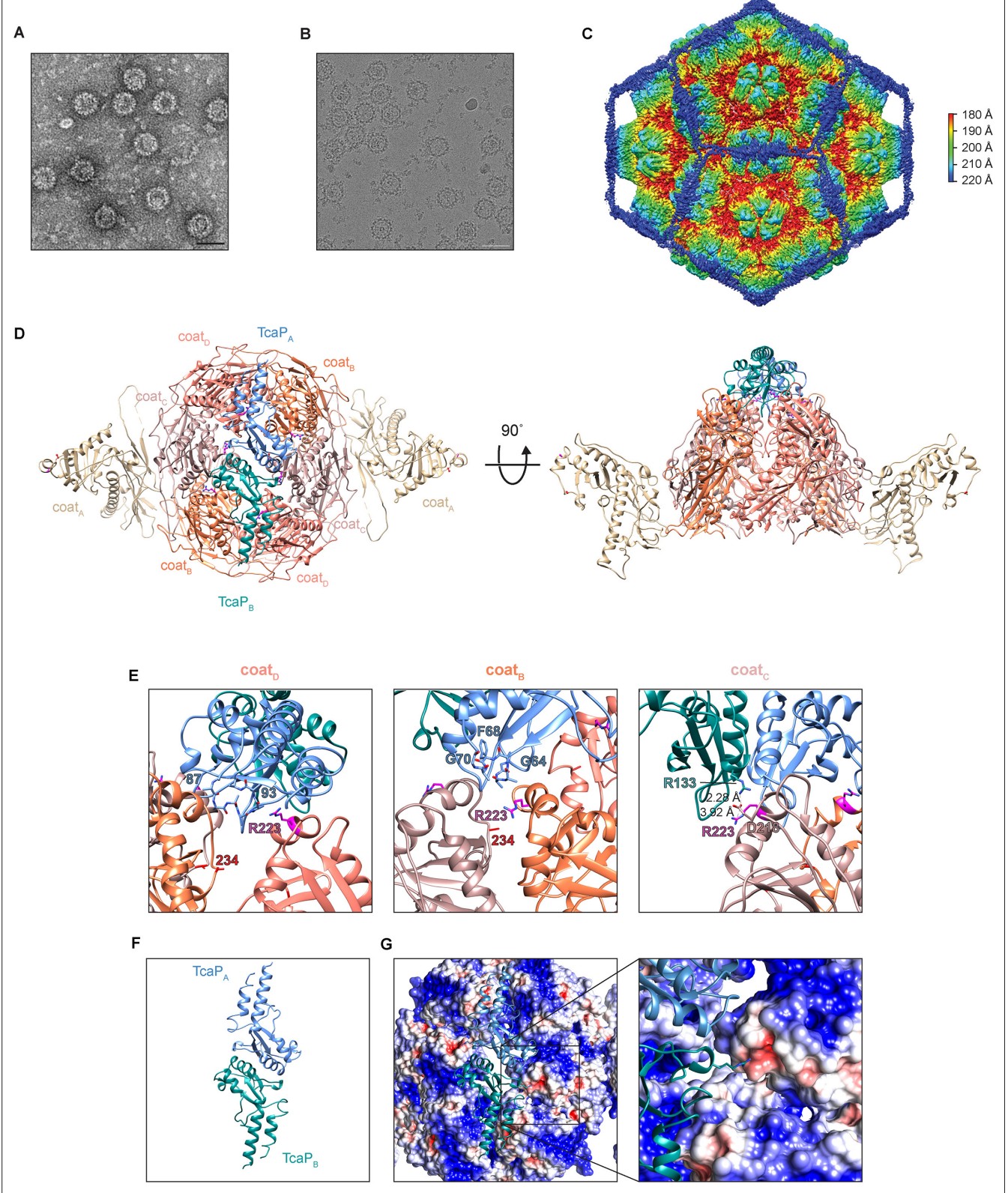

**Figure 4.** Cryogenic electron microscopy (cryo-EM) reveals TcaP is an external scaffold. (**A, B**) Representative micrographs from three independent biological replicates from (**A**) transmission electron and (**B**) cryo-electron microscopy of phage-inducible chromosomal island-like element (PLE) procapsid-like particle (PLPs) produced from co-expression of ICP1's coat and PLE's TcaP. Scale bars are 50 nm. (**C**) Isosurface reconstruction of PLE PLPs, resolved to 3.4 Å, colored radially (Å) as indicated in the legend. (**D**) Ribbon model of the solved structure for eight coat proteins, two from the

*Figure 4 continued on next page*

*Figure 4 continued*

adjacent pentamers (coat$_A$ tan) and six in a hexamer (coat$_B$ coral, coat$_C$ salmon, and coat$_D$ pink), and two partial TcaP proteins (TcaP$_A$ blue and TcaP$_B$ teal). Substituted residues in coat that escape TcaP-mediated remodeling are shown as balls and sticks (magenta for R223 and red for E234) and lie along the region of the hexamer where TcaP binds. Side chains that could not be fully resolved are modeled as alanines. (**E**) Details of the interactions between TcaP and the coat subunits. Residues in TcaP that contact coat are shown as sticks. For TcaP, the numbers, colored according to their chain, indicate the first and last residue within the region of contact. Distance measurements between residues are shown as dashed lines and measured in Å. Side chains that could not be fully resolved are modeled as alanines. (**F**) Ribbon model of the solved structure of the partial TcaP dimer, oriented as in panels (**D**) and (**G**). (**G**) Electrostatic potential surface representation of the coat with a ribbon diagram of TcaP show the negative pocket on coat$_C$ that is filled by the positively charged arginine from TcaP. Electrostatic potential is calculated using APBS and the coloring is from –5 kT/e (red) to +5 kT/e (blue).

The online version of this article includes the following source data and figure supplement(s) for figure 4:

**Source data 1.** Cryogenic electron microscopy (cryo-EM) data collection parameters used to create *Figure 4B–G*.

**Source data 2.** Data reconstruction model parameters used to create *Figure 4B–G*.

**Figure supplement 1.** Fourier shell correlation (FSC) curve.

**Figure supplement 2.** Comparison of TcaP and Sid structures in procapsid-like particles.

## TcaP is conserved in PLEs

Having established that PLE1's TcaP acts as an alternative external scaffold that directs the assembly of ICP1's coat protein into small capsids, we assessed the conservation of *tcaP* across genetically distinct PLEs. While PLE1 is the most well-studied PLE, there are nine additional PLEs that have been discovered to date (*Angermeyer et al., 2021*). Every PLE encodes a *tcaP* allele in the same locus, which encodes for proteins sharing 62–100% amino acid identity with TcaP$^{PLE1}$ (*Figure 5A*, *Figure 5—figure supplement 1A*). Largely, the regions that contact coat proteins in the PLP are conserved (*Figure 5—figure supplement 1C*). The conservation of this protein in the 10 PLEs suggests that all PLEs share the TcaP-mediated capsid remodeling strategy.

Curiously, the most divergent *tcaP* allele, *tcaP*$^{PLE5}$, is 237 nucleotides (~26%) shorter than *tcaP*$^{PLE1}$ due to an apparent truncation of the N-terminal coding region. However, a nucleotide alignment of the *tcaP*$^{PLE1}$ sequence with the *tcaP*$^{PLE5}$ 5′-UTR through the end of the coding sequence revealed two small deletions, one of which results in a frameshift and premature stop codon in the *tcaP*$^{PLE5}$ allele (*Figure 5A and B*). The alternative start site downstream of the deletions would restore the original reading frame for *tcaP*$^{PLE5}$ (*Figure 5B*). Given the conservation of full-length *tcaP* alleles in other PLEs, and the structural data highlighting interactions between the TcaP$^{PLE1}$ with ICP1's coat proteins, we assessed if the truncated TcaP$^{PLE5}$ was functional to redirect ICP1 virion assembly.

To start, we generated PLE particles from the wild type PLE5 strain carrying an empty vector and assessed their morphology by TEM. We found the PLE5 virions did not have small capsids, but rather they had large, ICP1-size capsids (*Figure 5C*), similar to virions from the PLE1Δ*tcaP* strain (*Figure 2B*). Consistent with this, when we directly addressed the scaffolding activity of the truncated allele by expressing it *in trans* during ICP1 infection and assessing plaque formation, we found that unlike p*tcaP*$^{PLE1}$, which inhibited ICP1 plaquing (*Figure 1E*), p*tcaP*$^{PLE5}$ did not inhibit ICP1 plaque formation (*Figure 5E*), supporting the conclusion that TcaP$^{PLE5}$ is not a functional scaffold. The reconstruction of the PLE1 PLPs (*Figure 4C*) revealed residues 63–70 in TcaP$^{PLE1}$ that specifically contact coat$_B$. That region is notably lacking from the truncated TcaP (*Figure 5—figure supplement 1C*). Further, we predict that a TcaP protein missing 79 amino acids would not be long enough to form the scaffolding cage around an assembling T = 4 procapsid explaining the lack of scaffolding activity for this TcaP variant.

Next, we addressed the transduction efficiency of PLE5 since the efficiency of PLE1's transduction decreased in the absence of TcaP (*Figure 2D*), demonstrating an advantage to packaging the PLE genome into smaller capsids. Previous work reported similar transduction efficiency for PLEs1-5 (*O'Hara et al., 2017*), and we recapitulated those results here, showing that PLE5 has only a subtle, approximately twofold decrease in transduction efficiency relative to PLE1 (*Figure 5F*). As this seemed in conflict with the PLE1Δ*tcaP* data, where there was over a tenfold decrease in PLE1 transduction without TcaP (*Figure 2D*), we directly examined the effect of small capsid production on PLE5 transduction by expressing TcaP$^{PLE1}$ *in trans*. First, we confirmed that p*tcaP*$^{PLE1}$ in PLE5(+) *V. cholerae* during ICP1 infection directed the assembly of small capsids (*Figure 5D*), which supported our hypothesis that the unmodified capsids produced by PLE5 are the result of this PLE encoding a nonfunctional

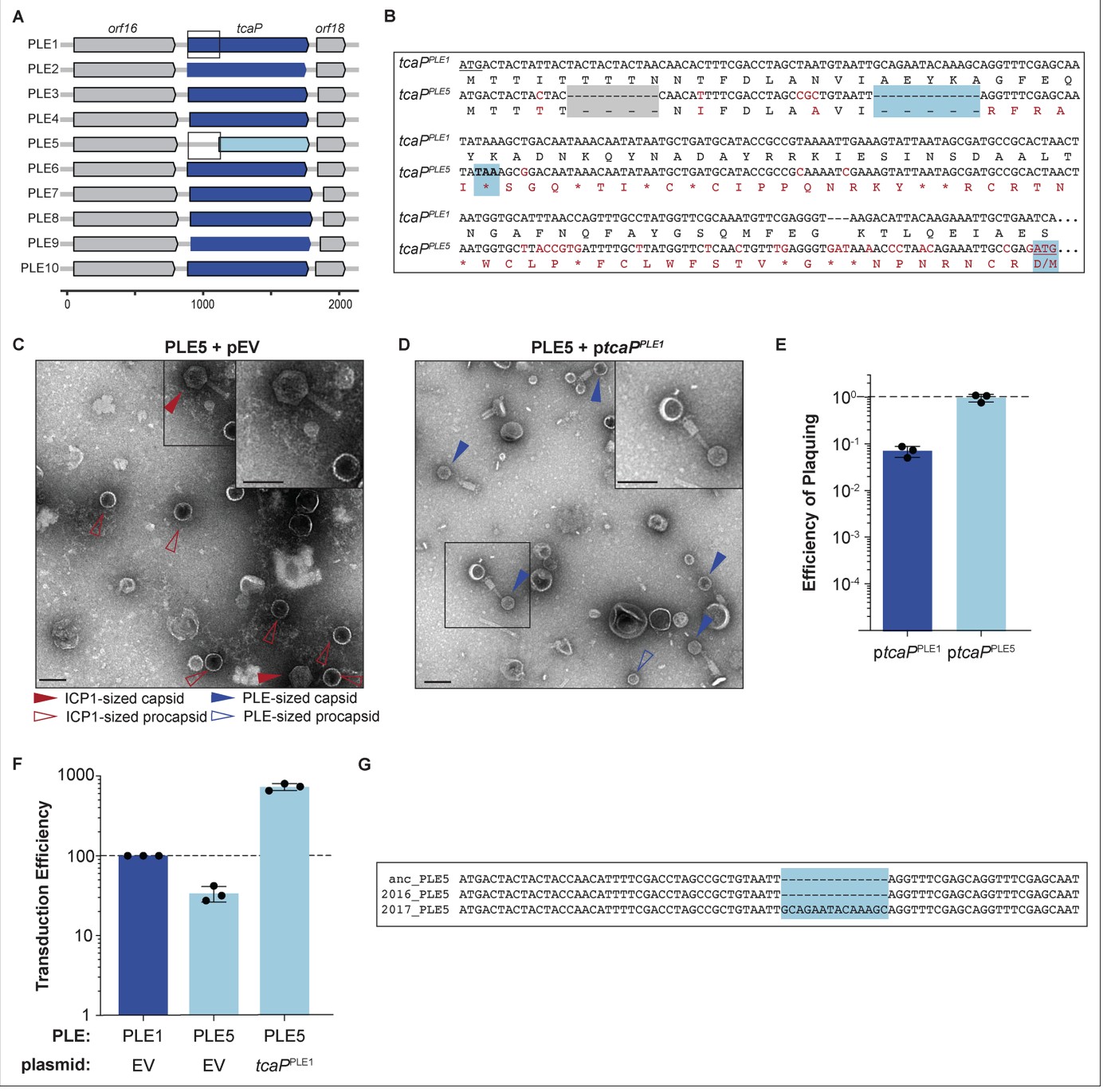

**Figure 5.** Genetic analysis of *tcaP* alleles from 10 phage-inducible chromosomal island-like element (PLEs) reveals PLE5's nonfunctional TcaP variant. (**A**) Gene graphs of *tcaP* and its two neighboring genes, from the 10 known unique PLEs. PLE5's *tcaP*, the shortest allele, is shown in light blue, while the other, full-length *tcaP* alleles are shown in dark blue, and the neighboring genes are shown in gray. Lower scale bar is in nucleotides. Boxes outline regions aligned in (**B**). (**B**) Alignment of the region encoding *tcaP* and their translated products from PLE1 and PLE5 from the boxed regions in panel (**A**). Nonidentical nucleotides and amino acids are shown in red on the PLE5 sequence, gaps are shown as dashes on either sequence, and stop codons are shown as asterisks. The gray box indicates an in-frame deletion. The blue boxes indicate the notable features of the *tcaP*^PLE5 sequence: the 14-nucleotide deletion that results in the frameshift, the resulting early stop codon, and the ATG and M of the alternative, originally annotated start site, which restores the original reading frame. (**C, D**) Representative transmission electron micrographs (TEMs) from 2 to 3 independent biological replicates of lysate from ICP1-infected strains of PLE5(+) *V. cholerae* (**C**) with an empty vector (pEV), or (**D**) expressing *tcaP* from PLE1 (p*tcaP*^PLE1). Scale bars are 100 nm. Arrowheads show capsids and their sizes according to the legend. (**E**) Efficiency of ICP1 plaquing on *V. cholerae* expressing *tcaP* from PLE1 (p*tcaP*^PLE1) or *tcaP* from PLE5 (p*tcaP*^PLE5) (from the originally annotated start site producing the truncated allele) compared to an empty vector (pEV). Each dot represents a biological replicate, bars represent the mean, and error bars show the standard deviation. The dotted line indicates an efficiency of

*Figure 5 continued on next page*

Figure 5 continued

plaquing of 1, where the expression of TcaP is not inhibitory to plaque formation. (**F**) Transduction efficiency of the strain indicated relative to PLE1 with an empty vector. Each dot represents a biological replicate, bars represent the mean, and error bars show standard deviation. The dotted line indicates an efficiency of 100%. (**G**) Alignment of the first nucleotides of the *tcaP* alleles from the PLE5 variants encoding the 'ancestral' (anc) sequences from before 1991, from 2016, or from 2017. The light blue box highlights the 14-nucleotide insertion in the PLE5 sequence from 2017.

The online version of this article includes the following source data and figure supplement(s) for figure 5:

**Source data 1.** Data used to create *Figure 5E*.

**Source data 2.** Data used to create *Figure 5F*.

**Source data 3.** Uncropped micrographs used to create *Figure 5C and D*.

**Figure supplement 1.** All phage-inducible chromosomal island-like element (PLEs) encode *tcaP* alleles.

TcaP allele. Next, we found that PLE5 transduction increased with p*tcaP*^PLE1^, exceeding the transduction efficiency of PLE5 and PLE1 (*Figure 5F*). This result suggests that the capacity for transduction is different for PLE1 and PLE5, but still nonetheless demonstrates that small capsids promote PLE transduction.

Since both PLE1 and PLE5 transduction efficiency was higher when small capsids were made, this raised the question of PLE5's maintenance in the *V. cholerae* population while lacking a functional scaffold. A recent analysis of sequenced *V. cholerae* isolates outlined a pattern where a PLE variant emerges, often rises to dominance, then is replaced by a different variant (*Angermeyer et al., 2021*). This pattern revealed that after its dominance from before 1960 until its disappearance in 1991 (for which there are 21 *V. cholerae* isolates harboring identical PLE5 sequences), PLE5 re-emerged in 2016 and 2017. Genetic drift within a given PLE variant appears to be very rare: a particular PLE variant is typically 100% conserved at the nucleotide level across the entire mobile genetic element despite residing in different isolates of *V. cholerae* (*O'Hara et al., 2017*; *McKitterick et al., 2019*). However, on rare occasions, single-nucleotide polymorphisms between the same PLE variant are found in different *V. cholerae* isolates (*Barth et al., 2021*). Strikingly, the *tcaP*^PLE5^ sequence from 2017 showed a 14-nucleotide insertion which restored a full-length *tcaP* allele (*Figure 5G*, *Figure 5—figure supplement 1B*). The re-establishment of the full-length *tcaP* allele in the contemporary PLE5, along with the preservation of full-length alleles in other PLEs, supports conservation of TcaP for its role in capsid remodeling to promote PLE transduction and inhibit ICP1.

## Discussion

In this work, we characterized the mechanism of capsid remodeling in the phage-satellite system ICP1-PLE. We identified a PLE-encoded scaffold, TcaP, and demonstrated its necessity and sufficiency to redirect the assembly of ICP1's coat proteins into small capsids. Using a heterologous assembly system and cryo-EM, we discovered that TcaP functions as an external scaffold that assembles into a cage-like structure around the procapsid, favoring the smaller T = 4 morphology (*Figure 6A*). We found that TcaP is conserved in all PLEs. Further, TcaP-mediated small capsid assembly is advantageous to PLE as the production of small capsids leads to more efficient transduction of PLE's genome, a unique feature not seen in other headful packaging capsid-remodeling satellites. The data presented here further highlight similarities between PLE and other satellites while underscoring features that make PLE unique.

Initially, TcaP's function was unknown since homology-based and structural prediction searches were of low confidence and failed to identify known domains. However, the structure of PLE PLPs showed molecular details of the TcaP-coat interactions and revealed a striking resemblance to that of PLPs formed by the *E. coli* phage P2 coat protein (GpN), scaffold (GpO), and satellite P4 external scaffold (Sid) (*Kizziah et al., 2020*; *Figure 4—figure supplement 2*). Despite the low sequence conservation between the ICP1 and P2 coat proteins (20% identity), both have an HK97-like fold and are assembled into T = 4 icosahedral procapsids measuring ~45 nm in diameter in the presence of their satellite-encoded external scaffolds. TcaP and Sid also share ~20% amino acid identity; however, TcaP is 53 amino acids longer than Sid. TcaP, like Sid, assembles into long filaments that form dimers that span the twofold axes and trimers of dimers meeting at the threefold axes of the procapsid. Given how TcaP and Sid interact with themselves to form dimers and trimers wherein hexamers are linked to each

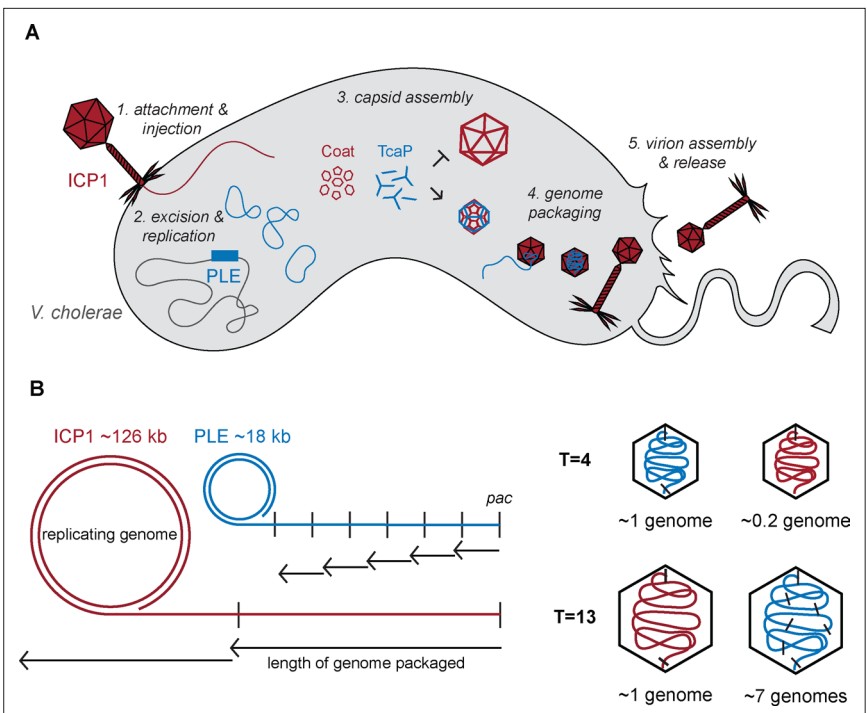

**Figure 6.** Model of TcaP-mediated small capsid assembly. (**A**) A cartoon model of ICP1 infecting a PLE(+) *V. cholerae* cell. Injection of ICP1's genome triggers activation of phage-inducible chromosomal island-like element (PLE), then TcaP directs the assembly of coat proteins into small capsids, inhibiting the formation of large capsids. PLE's genome is then packaged into the small capsids, a process that likely triggers the removal of the external scaffold, TcaP. PLE virions are released from the cell. ICP1 components are red, PLE components are blue, and *V. cholerae* components are grey. (**B**) A model showing the impact of capsid size on genome packaging. The replicated ICP1 and PLE genomes form concatemers, from one *pac* site (indicated as a vertical line) to the next. Headful packaging results in ~105–110% of the genome within the capsid. The length of the genome packaged is indicated by the small arrows for T = 4 capsids and longer arrows for T = 13 capsids. ICP1's genome is partially packaged into a T = 4 capsid and several copies of PLE's genome are packaged into a T = 13 capsid. ICP1's genome is red and PLE's is blue.

other by the length of TcaP or Sid dimers, it is not surprising how the resulting scaffold cage accommodates only procapsids with T = 4 symmetry. Additional hexamers would not be accommodated within the cage formed by the external scaffolds, and thus the capsid is blocked from forming higher T number structures. Despite their similarities, TcaP and Sid differ in their specific interactions with the coat proteins. Unlike Sid, which only contacts two proteins in the hexamer (both GpN$_B$ subunits), both TcaP monomers form interactions with three coat subunits, thus contacting all six coat subunits. TcaP and Sid also differ in their tertiary structures (*Figure 4—figure supplement 2*), further demonstrating the convergent evolution of the external scaffold capsid-remodeling strategy in divergent satellites.

Few external scaffolds have been described in viruses, and thus there are open questions about their role in capsid assembly, especially in relation to internal scaffolds. Outside of P4 and PLE, the only other identified external scaffolds are those in the ssDNA phages of the Microviridae family. A representative phage, ΦX174, requires an external scaffold (protein D) to form procapsids (*Dokland et al., 1997*; *Hafenstein and Fane, 2002*); however, the structure of this protein is different from TcaP and Sid (*Cherwa et al., 2011*). Unlike the trimers formed by the satellite scaffolds, protein D seems to form tetramers in solution and directs the assembly of the coat proteins into an ~27-nm-wide procapsid, despite the lack of direct coat–scaffold interactions. Notably, the small ΦX174 capsids still require an internal scaffold for assembly. Similarly, P4 is dependent on P2's internal scaffold (GpO) to form viable progeny virions, likely because the internal scaffold is important for the incorporation of the portal (*Chang et al., 2008*). Curiously, SaPIbov1, a satellite which uses an alternative internal scaffold to direct assembly of small capsids, also requires 80α's internal scaffold (Gp46) for viable satellite virions (*Spilman et al., 2012*), likely for a similar reason. PLEs do not appear to encode portal

proteins and thus we anticipate that PLE would similarly be dependent on ICP1 portal incorporation for progeny production. In the heterologous assembly assay, we show that TcaP's scaffolding activity is dominant over ICP1's scaffold, such that the resulting particles are satellite-sized in the presence of both scaffolds. These data suggest that if PLE depended on ICP1's scaffold for portal incorporation, TcaP's scaffolding activity would still be sufficient to direct the assembly of small capsids. As the portal was not included in the PLP assays here, it will be useful for future studies to directly address this requirement and to assess the complete protein makeup of PLE procapsids produced during ICP1 infection of PLE(+) *V. cholerae* to validate what PLE and ICP1 components comprise the procapsids.

Like ICP1 escaping TcaP activity, P2 can escape Sid-mediated capsid remodeling through point substitutions in the coat (*Kizziah et al., 2020*). Both of the two escape substitutions we identified in ICP1 lies in the A-domain, in the center of the hexamer across the twofold axis of symmetry where TcaP binds and one residue directly contacts TcaP. The A-domain has been implicated in regulating capsid size and assembly in many viral systems (reviewed in *Suhanovsky and Teschke, 2015*). Similarly, the five suppressor substitutions that have been identified in P2's coat protein lie in the hexamer along Sid's binding site, but they lie slightly outside of the A-domain and are instead at the end of the P-loop. Three substitutions are in residues that directly contact Sid, like R223 in TcaP. The other two residues in P2's coat likely function through other means to disrupt the interaction (*Kizziah et al., 2020*), as is probably the case for the second substitution seen in ICP1, E234K. Curiously, the PLP assembly data showed that the coat$^{R223H}$ only partially escaped TcaP (*Figure 3*) while this substitution was sufficient in vivo to escape TcaP's inhibition of plaque formation (*Figure 1E*). These inconsistencies can likely be explained by the lack of the additional components that comprise PLE native procapsids in our assembly platform, as well as the relative protein concentrations and dynamic regulation that occurs during infection. However, despite the selection for phages that escape TcaP-mediated remodeling in the laboratory, no mutations in the A-domain exist in the current collection of 67 sequenced isolates of ICP1. Perhaps the mutations are not selected for in nature because the biophysical properties of these coat variants are incompatible with assembly in the natural conditions of the human gut or in estuaries where *V. cholerae* and ICP1 reside. In line with this hypothesis, one of the coat mutants we tested, coat$^{E234K}$, was not assembly competent in the heterologous assembly system (*Figure 3—figure supplement 4*), likely due to its predicted role in stabilizing the coat protein. Moreover, PLE is severely inhibitory to ICP1, employing several redundant strategies that independently restrict phage production, and thus *tcaP* is dispensable for phage inhibition (*Hays and Seed, 2020*). Instead of selecting for mutations that individually escape PLE's inhibitory proteins, ICP1 encodes broader strategies that degrade the PLE genome, such as the phage-encoded CRISPR-Cas system (*Seed et al., 2013*; *McKitterick et al., 2019*), Odn (*Barth et al., 2021*), or Adi (*Nguyen et al., 2022*).

The inhibitory effects on the helper phage resulting from the production of small capsids from hijacked coat proteins are comparable between PLEs and other satellites that depend on helper coat proteins for virion production (*Damle et al., 2012*; *Carpena et al., 2016*; *Kim et al., 2001*). PLE, like other satellites, has a genome size of ~15–20 kb, which is compatible with the satellite-modified T = 4 capsid size. However, the ICP1 genome does not follow the pattern seen in other helper phages, which typically are ~30–40 kb and packaged into similarly sized capsids (icosahedral T = 7 or prolate $T_{end}$ = 4, $T_{mid}$ = 14). ICP1 has a large ~125 kb genome packaged into a T = 13 capsid. Therefore, PLE and ICP1 have the most dramatic size difference between satellite and helper phage. The consistent level of inhibition is likely explained by the fact that regardless of how much smaller the capsid is, packaging anything less than the full length of the genome results in non-infectious virions. However, the effect of capsid size on the satellite packaging can be different. Phages and satellites that use *cos* site packaging strategies specifically cleave the genome only at those sites and thus package unit lengths of their genomes into a capsid. For *cos* packaging satellites that depend on helper phage coat proteins, if their genome is proportional to two or three lengths of their helper phages, they can package their genome into helper-sized capsids with no or reportedly little reduction in efficiency; however, genomes that are not proportional suffer from slight reductions in transduction (*Shore et al., 1978*; *Carpena et al., 2016*). On the other hand, with headful packaging systems, which have been predicted for ICP1 and PLE (*Barth et al., 2020*), replicated genome concatemers will be threaded into the capsid until the capsid is full, signaling the terminase to stop packaging and cut the genome. Most phage's capsids accommodate 103–110% of their genome length (*Rao and Feiss, 2015*). While the SaPIs that use headful packaging are unaffected by packaging their genomes into large capsids,

PLE's transduction is severely reduced when packaged into ICP1-size capsids. The difference in capsid size between the SaPI helper, 80α (T = 7), and PLE's helper, ICP1 (T = 13), explains this result. Approximately two SaPI genomes would need to be packaged to fill the larger, T = 7 capsid produced in the absence of its remodeling proteins CpmB and CpmA. Meanwhile, 6–7 PLE genomes would need to be packaged into ICP1's large, T = 13 capsids in the absence of TcaP (*Figure 6B*). This larger difference in genome copies per capsid supports why PLE is the only satellite to suffer a transduction defect in the absence of capsid remodeling. The negative effect on PLE transduction in the absence of capsid remodeling suggests that PLEs benefit from TcaP's activity, explaining why it is largely conserved in all PLEs discovered to date.

By packaging each of its replicated genomes into capsids proportional to its size, PLE makes more transducing particles and increases the odds of horizontal transfer of its genome. Theoretically, high transduction efficiency would be advantageous and selected for in PLEs. If this is the case, the expectation would be that all PLEs encode a functional TcaP that directs the assembly of small capsids. However, older isolates of PLE5 lack such an allele and consequently package their genomes into large capsids (*Figure 5C*). If the loss of TcaP decreased PLE5's fitness by decreasing its ability to spread horizontally, it could be expected that this PLE would quickly be replaced by more fit PLEs in the population. Surveillance data show that PLE5 was detected in *V. cholerae* genomes from as early as 1931 until 1991 and again in 2016 and 2017 (*Angermeyer et al., 2021*), suggesting PLE5 was maintained in the population for relatively long periods of time. Curiously, in an isogeneic background and under lab conditions, PLE5 transduces nearly as efficiently as PLE1 (*O'Hara et al., 2017*), though transduction efficiency is increased by the formation of small capsids (*Figure 5D and F*). These data suggest that PLE5 encodes other factors that contribute to its transduction and promote its maintenance in the population. It is unclear at what frequency and/or efficiency PLE transduction occurs in native conditions in estuaries or in the human intestinal tract where ICP1 preys on *V. cholerae*. It may be that low levels of transduction are sufficient to maintain a PLE in the population. Alternatively, PLEs could be maintained primarily through vertical transmission (*Angermeyer et al., 2021*). The arsenal of anti-ICP1 factors encoded by PLE, even in the absence of TcaP activity, seem beneficial for *V. cholerae*, supporting vertical transmission of PLE5 as a means for its maintenance in the population. However, the emergence of a restored full-length allele of *tcaP* in PLE5 in 2017, paired with the conservation of TcaP in other PLEs and evidence of horizontal PLE transfer in some clinical isolates (*Angermeyer et al., 2021*), suggests capsid remodeling is advantageous due to its role in transduction and is selected for in PLEs in nature.

PLE is a unique satellite and its helper phage ICP1 is also dissimilar from other helper phages (*de Sousa et al., 2022*). In addition to its large capsid size, ICP1 is different from other documented helper phages in that it is an obligate lytic phage. This raises questions about whether the characteristics of the ICP1-PLE parasitism are specifically due to the lytic nature of ICP1. For example, PLE's anti-phage activity is the most potent of any satellite where ICP1 progeny production is completely blocked by PLE. Perhaps this more severe phage restriction is important for the protection of neighboring *V. cholerae* cells from lethal infection. In the case of temperate phages, the presence of many phage kin may promote lysogeny, and the infected cell would survive. As this is not a possibility for *V. cholerae* infected by ICP1, PLE's reduction of ICP1 in the environment may be a necessary means of protection for the bacterial population. In support of the selection for phage inhibition, PLEs encode multiple redundant mechanisms to block ICP1. It will be interesting to see if other lytic phages are parasitized by satellites and if there is conservation of distinct inhibitory mechanisms. Recent work has highlighted distantly related elements in other *Vibrio* species that may be elements with similarities to PLEs (*LeGault et al., 2022*), but putative helper phages have not been identified. A recent study described a density separation and sequencing technique to identify novel satellites (*Eppley et al., 2022*) that may help illuminate the diversity in phage-satellite pairings. As this work has highlighted, the convergent evolution of satellite strategies to hijack aspects of a helper phage's lifecycle may not be obvious at the sequence level, and the characterization of novel satellites can reveal features that unite or separate different families of satellites.

## Materials and methods

**Key resources table**

| Reagent type (species) or resource | Designation | Source or reference | Identifiers | Additional information |
|---|---|---|---|---|
| Genetic reagent (*V. cholerae*) | p*tcaP*[PLE1] | This paper | CMB2 | *V. cholerae* E7946 pKL06 $P_{tac}$RiboE[4]-*tcaP* (PLE1); strepR[1], cmR[2] |
| Genetic reagent (*V. cholerae*) | pEV[5] | This paper | CMB3 | *V. cholerae* E7946 pKL06 $P_{tac}$ RiboE-Empty Vector; strepR, cmR |
| Genetic reagent (*V. cholerae*) | p*tcaP*[PLE5] | This paper | CMB41 | *V. cholerae* E7946 pKL06 $P_{tac}$ RiboE-*tcaP* (PLE5); strepR, cmR |
| Genetic reagent (*V. cholerae*) | PLE1Δ*tcaP* pEV | This paper | CMB71 | *V. cholerae* E7946 PLE1::*kanR*Δ*tcaP* pKL06 $P_{tac}$RiboE-EV; strepR, cmR, kanR[3] |
| Genetic reagent (*V. cholerae*) | PLE1Δ*tcaP* p*tcaP*[PLE1] | This paper | CMB73 | *V. cholerae* E7946 PLE1::*kanR*Δ*tcaP* pKL06 $P_{tac}$RiboE-*tcaP* (PLE1); strepR, cmR, kanR |
| Genetic reagent (*Escherichia coli*) | p*coat*::6xHis, pEV | This paper | CMB548 | *E. coli* BL21 pETDUET[6] $P_{tac}$-*gp122*::6xHis, $P_{tac}$-EV |
| Genetic reagent (*E. coli*) | p*coat*::6xHis, p*tcaP* | This paper | CMB555 | *E. coli* BL21 pETDUET $P_{tac}$-*gp122*::6xHis, $P_{tac}$-*tcaP* |
| Genetic reagent (*E. coli*) | p*coat*::6xHis, p*scaffold* | This paper | CMB560 | *E. coli* BL21 pETDUET $P_{tac}$-*gp122*::6xHis, $P_{tac}$-*gp124* |
| Genetic reagent (*E. coli*) | p*scaffold-decoration-coat*::6xHis, p*tcaP* | This paper | CMB562 | *E. coli* BL21 pETDUET $P_{tac}$-*gp124-gp123-gp122*::6xHis, $P_{tac}$-*tcaP* |
| Genetic reagent (*E. coli*) | p*scaffold-decoration-coat*::6xHis, pEV | This paper | CMB568 | *E. coli* BL21 pETDUET $P_{tac}$-*gp124-gp123-gp122*::6xHis, $P_{tac}$-*tcaP* |
| Genetic reagent (*V. cholerae*) | PLE1 pEV | This paper | CMB577 | *V. cholerae* E7946 PLE1::*kanR* pKL06.2 $P_{tac}$RiboE-EV; strepR, cmR, kanR[3] |
| Genetic reagent (*V. cholerae*) | PLE5 pEV | This paper | CMB579 | *V. cholerae* E7946 PLE5::*kanR* pKL06.2 $P_{tac}$RiboE-EV; strepR, cmR, kanR[3] |
| Genetic reagent (*V. cholerae*) | PLE5 p*tcaP*[PLE1] | This paper | CMB581 | *V. cholerae* E7946 PLE5::*kanR* pKL06.2 $P_{tac}$RiboE-*tcaP*(PLE1); strepR, cmR, kanR[3] |
| Genetic reagent (*E. coli*) | p*scaffold-decoration-coat*[R223H]::6xHis, pEV | This paper | CMB603 | *E. coli* BL21 pETDUET $P_{tac}$-*gp124-gp123-gp122*[R223H]::6xHis, $P_{tac}$-EV |
| Genetic reagent (*E. coli*) | p*scaffold-decoration-coat*[R223H]::6xHis, p*tcaP* | This paper | CMB604 | *E. coli* BL21 pETDUET $P_{tac}$-*gp124-gp123-gp122*[R223H]::6xHis, $P_{tac}$-*tcaP* |
| Genetic reagent (*E. coli*) | p*coat*[R223H]::6xHis, pEV | This paper | CMB605 | *E. coli* BL21 pETDUET $P_{tac}$-*gp122*[R223H]::6xHis, $P_{tac}$-EV |
| Genetic reagent (*E. coli*) | p*coat*[R223H]::6xHis, p*tcaP* | This paper | CMB606 | *E. coli* BL21 pETDUET $P_{tac}$-*gp122*[R223H]::6xHis, $P_{tac}$-*tcaP* |
| Genetic reagent (*E. coli*) | p*coat*::6xHis, p*tcaP*; p*coat* | This paper | CMB609 | *E. coli* BL21 pETDUET $P_{tac}$-*gp122*::6xHis, $P_{tac}$-*tcaP* and pCDFDuet $P_{tac}$-*gp122* |
| Genetic reagent (*E. coli*) | p*coat*[E234K]::6xHis, pEV | This paper | CMB574 | *E. coli* BL21 pETDUET $P_{tac}$-*gp122*[E234K]::6xHis, $P_{tac}$-EV |
| Genetic reagent (*E. coli*) | p*coat*[E234K]::6xHis, p*tcaP* | This paper | CMB575 | *E. coli* BL21 pETDUET $P_{tac}$-*gp122*[E234K]::6xHis, $P_{tac}$-*tcaP* |
| Genetic reagent (*E. coli*) | p*coat*[E234K]::6xHis, p*scaffold* | This paper | CMB576 | *E. coli* BL21 pETDUET $P_{tac}$-*gp122*[E234K]::6xHis, $P_{tac}$-*gp124* |
| Genetic reagent (bacteriophage ICP1) | ICP1 wild type | *McKitterick et al., 2019* | KSΦ49 | ICP1_2011_Dha_A Δspacer 9 (accession: MH310933) |
| Genetic reagent (bacteriophage ICP1) | ICP1 *coat*[R223H] | This paper | Phage 1, DDΦ78 | ICP1_2011_Dha_A Δspacer 9 Gp122[R223H] |
| Genetic reagent (bacteriophage ICP1) | ICP1 *coat*[E234K] | This paper | Phage 2, DDΦ79 | ICP1_2011_Dha_A Δspacer 9 Gp122[E234K] |
| Gene (bacteriophage ICP1) MH310933 | *gp122; coat* | N/A | Locus tag: ICP12011A_121 protein accession: AXY82215.1 | Major capsid protein |

| Reagent type (species) or resource | Designation | Source or reference | Identifiers | Additional information |
|---|---|---|---|---|
| Gene (bacteriophage ICP1) MH310933 | *gp123; decoration* | N/A | Locus tag: ICP12011A_122 protein accession: AXY82216.1 | Capsid decoration protein |
| Gene (bacteriophage ICP1) MH310933 | *gp124; scaffold* | N/A | Locus tag: ICP12011A_123 protein accession: AXY82217.1 | Capsid assembly scaffolding protein |
| Gene (*V. cholerae*) KC152960.1 | *orf17; tcaP (PLE1)* | N/A | Locus tag: orf17 protein accession: AGG09411.1 | Satellite external scaffolding protein |
| Gene (*V. cholerae*) CP001236.1 | *orf17; tcaP (PLE5_2008)* | N/A | Locus tag: VC395_A0477 Protein accession: ACP11313.1 | Satellite external scaffolding protein |
| Gene (*V. cholerae*) CP045718.1 | *orf17; tcaP (PLE5_2016)* | N/A | Locus tag: GG844_03730 Protein accession: QGF30303.1 | Satellite external scaffolding protein |
| Gene (*V. cholerae*) Assembly: GCF_007050395.1 | *orf17; tcaP (PLE5_2017)* | N/A | Locus tag: DM782_RS18270 Protein accession: WP_001912195.1 | Satellite external scaffolding protein |
| Commercial assay or kit | NEBNext Ultra II DNA Library preparation kit | New England Biolabs | NEB #E7645, E7103 | |
| Commercial assay or kit | DNeasy blood and tissue DNA purification kit | QIAGEN | QIAGEN 69506 | |
| Software, algorithm | cryoSPARC | *Punjani et al., 2017* | cryoSPARC v4.0.3 | |
| Software, algorithm | RELION | *Kimanius et al., 2021* | RELION 4.0 | |
| Software, algorithm | AlphaFold2 | *Jumper et al., 2021* | AlphaFold2 | |
| Software, algorithm | *Phenix* | *Liebschner et al., 2019* | *Phenix* | |
| Software, algorithm | COOT | *Emsley et al., 2010* | COOT | |
| Software, algorithm | Pyem | *Asarnow et al., 2019* | Pyem | |

## Bacterial growth conditions

*V. cholerae* and *E. coli* were propagated at 37°C on LB agar or in LB broth with aeration. Where needed, antibiotics were used at the following concentrations: streptomycin 100 µg/mL, kanamycin 75 µg/mL, spectinomycin 100 µg/mL, chloramphenicol 2.5 µg/mL (solid media) or 1.25 µg/mL (liquid media) for *V. cholerae* and 25 µg/mL for *E. coli*, and carbenicillin 50 µg/mL.

## Strain construction

*V. cholerae* carrying PLE1 marked with a kanamycin resistance cassette downstream of the last ORF (*O'Hara et al., 2017*) was made naturally competent and transformed with DNA fragments containing a spectinomycin resistance marker flanked by frt recombinase sites assembled to up and downstream regions of homology by splicing by overlap extension PCR as previously described (*Dalia et al., 2014*). Spectinomycin-resistant transformants were transformed with a plasmid carrying an FLP recombinase which was induced by 1 mM isopropyl-β-D-thiogalactopyranoside (IPTG) (Fischer BioReagents, 367-93-1) and 1.5 mM theophylline (Sigma, T1633-100G), allowing for the removal of the spectinomycin resistance cassette via recombination, resulting in an in-frame deletion. The strains were cured of the plasmid, and deletions were confirmed by PCR and Sanger sequencing.

Plasmid constructs were assembled using Gibson Assembly and transformed into *E. coli* BL21 or mated into *V. cholerae* via *E. coli* S17. A list of strains used in this study can be found in the Key resources table. Plasmids in *V. cholerae* have a P$_{tac}$ promoter downstream of a theophylline sensitive riboswitch. A list of oligos used in this study can be found in *Supplementary file 3*.

## ICP1 plaque assays

Overnight cultures of *V. cholerae* were diluted to OD$_{600}$ = 0.05 and grown in LB (supplemented with antibiotics where appropriate) with aeration at 37°C either directly to OD$_{600}$ = 0.3 or to OD$_{600}$ = 0.2 then induced with 1 mM IPTG and 1.5 mM theophylline for 20 min to reach OD$_{600}$ = 0.3, then mixed with pre-diluted phage samples. Phage attachment was allowed for 7–10 min prior to plating in 0.5% molten top agar (supplemented with antibiotics and inducer where appropriate) followed by overnight incubation at 37°C. Resulting individual plaques were counted.

## ICP1 mutant purification and whole-genome sequencing

*V. cholerae* carrying a plasmid encoding *tcaP* was used in a plaque assay as described above. As a control, ICP1_2011_Dha_A was used to infect *V. cholerae* expressing an empty vector. Plaques that formed on the *tcaP*-expressing strain were picked and purified on that strain two more times by plaque assay. High-titer phage stocks were prepared by sodium chloride (1 mM) polyethylene glycol 8000 (10%) precipitation or by centrifugation (26,000 × *g* for 90 min) and stored in STE (5 mM Tris pH 8.0, 100 mM NaCl, 1 mM EDTA). Prior to collecting genomic DNA (gDNA), phage stocks were treated with DNase for 30 min at 37°C to remove non-encapsidated DNA, then the enzyme was heat inactivated. gDNA was collected using a QIAGEN DNeasy blood and tissue DNA purification kit (QIAGEN, 69506) according to the manufacturer's protocols and genomic libraries were prepared for Illumina sequencing using the NEBNext Ultra II DNA Library preparation kit (NEB #E7645, E7103) as described in the manufacturer's protocols. Using an Illumina HiSeq4000 (University of California, Berkeley QB3 Core Facility), samples were sequenced by paired-end (2 × 150 bp). The genomes were assembled using SPAdes and analyzed using BreSeq (v0.33).

## Virion production for TEM and/or transduction

50 mL cultures of *V. cholerae* strains carrying plasmids were grown and induced as described above. ICP1 was added at a multiplicity of infection (MOI) of 2.5 and cultures were incubated until lysis (30–90 min). 1 mL of lysate was used for transduction assays (see below). Remaining lysates were concentrated by centrifugation at 26,000 × *g* for 90 min, resuspended in Phage Buffer 2.0 (50 mM Tris–HCl, 100 mM NaCl, 10 mM MgSO$_4$, 1 mM CaCl$_2$) overnight, treated 1:1 with chloroform (Fisher Scientific, C606-1) for 15 min, and centrifuged at 5000 × *g* for 15 min. The aqueous layer was collected and 5 μL was applied to a grid for TEM.

PLE transduction assays were carried out as previously described (*Netter et al., 2021*; *O'Hara et al., 2017*). Briefly, 1 mL of lysates from strains carrying PLE marked with a kanamycin resistance cassette downstream of the last ORF were treated with 10 μL chloroform, which was removed, along with bacterial debris, by centrifugation at 5000 × *g* for 15 min. The supernatant was collected and mixed 10:100 with a saturated overnight culture of spectinomycin resistant recipient *V. cholerae* cells (Δ*lacZ::spec^R*) supplemented with 10 mM MgSO$_4$ immediately prior to transduction. Recipients were incubated with lysates for 20 min at 37°C with shaking (220 rpm) and then serially tenfold diluted. The resulting dilutions were plated on LB agar plates supplemented with spectinomycin and kanamycin. A colony represents one PLE virion/transducing unit.

## Transmission electron microscopy

For the preparation of grids, 5 μL samples were incubated on a copper mesh grid (Formvar/Carbon 300, Electron Microscopy Sciences) for 60 s, wicked, immediately washed with sterile ddH$_2$O for 15 s, wicked, immediately stained with 1% uranyl acetate (Electron Microscopy Sciences, 22400-1) for 30 s, wicked and allowed to dry completely. Micrographs were collected with a FEI Tecnai-12 electron microscope operating at 120 kV.

## Production and purification of PLPs

Using Gibson Cloning, the coat gene fused to 6xHis was inserted into the pETDUET vector or the untagged coat was cloned into pCDFDuet. Either the ICP1 or PLE scaffold proteins were similarly

inserted via Gibson Assembly downstream of a second T7 promoter. In other variations of this plasmid, the genes were cloned with the intergenic sequences found in ICP1_2011_Dha_A. The pETDUET constructs were transformed into *E. coli* BL21 and grown on LB agar or in LB broth supplemented with carbenicillin (50 µg/mL). To reduce aggregation of PLE PLPs for cryo-EM, a pETDUET construct encoding *gp122*::6xhis and *tcaP* was co-transformed with a pCDFDuet construct expressing *gp122* (untagged), grown on LB agar or in LB broth supplemented with carbenicillin (50 µg/mL) and streptomycin (100 µg/mL). For protein production, overnight cultures were diluted 1:100 in 0.5–1 L, grown to $OD_{600}$ = 0.2–0.4, induced with 1 mM IPTG for 3–5 hr, and collected by centrifugation at 4000 × *g* for 20 min at 4°C. Pellets were then resuspended in 10% of the volume of Phage Purification Buffer (Phage Buffer 2.0 supplemented with 20 mM imidazole, 1 mM BME, and protease and phosphatase inhibitors [Cat# A32961/A32965, Pierce]), and frozen at –80°C. For protein purification, frozen samples were thawed, sonicated, and centrifuged at 12,000 × *g* for 60–90 min to remove membranes and debris. The particles were then pelleted at 26,000 × *g* for 90 min. The pellet was nutated overnight in Phage Purification Buffer, and the nutate was loaded onto a column packed with HisPur nickel-nitrilotriacetic acid (Ni-NTA) resin (Cat# PI88222, Thermo Scientific) pre-equilibrated with ice-cold Phage Purification Buffer. After two passes over the column, the flow-through was collected and the column was washed with wash buffer (Phage Buffer 2.0 supplemented with 50 mM imidazole, 1 mM BME, and 10% glycerol). Proteins were then eluted with one column volume of Elution Buffers 1–3 (Phage Buffer 2.0 supplemented with 1 mM BME and 10% glycerol and 150, 250, or 350 mM imidazole) and a final six column volume elution with Elution Buffer 4 (Phage Buffer 2.0 supplemented with 1 mM BME and 10% glycerol and 500 mM imidazole). For all experiments with coat[R223H] and the noted replicates of wild type coat, decoration, scaffold, and TcaP, protease and phosphatase inhibitors were added to all purification and elution buffers. Aliquots of the fractions were either boiled in Laemmli buffer and assessed by SDS-PAGE stained with Coomassie or applied to a grid and imaged by TEM.

Cesium chloride-based purification following affinity purification was carried out as follows. PLE PLP (coat::6xHis +TcaP) samples from the final elutions from affinity chromatography were concentrated on a 100K MWCO Amicon ultra filter (Cat# UFC810024, Millipore) to a final volume of ~1 mL. 1 mL steps of 1.6 g/cm³, 1.5 g/cm³, 1.4 g/cm³, and 1.3 g/cm³ CsCl were made in Phage Buffer 2.0 (50 mM Tris–HCl, 100 mM NaCl, 10 mM $MgSO_4$, 1 mM $CaCl_2$ pH 8.0), filter sterilized, and layered in ultracentrifuge tubes (thinwalled WX 5 mL, Cat# 1131, Thermo Scientific) then topped with 500 µL of PLE PLP sample. Samples were spun for 2 hr at 36,000 rpm (~110,000 × *g*) at 18°C (AH-650 swinging bucket rotor Thermo Scientific), then 250 µL fractions were manually collected. The protein content of fractions was assessed by SDS-PAGE/Coomassie. Fractions containing the most pure coat::6xHis and TcaP complexes were pooled, dialyzed against Phage Buffer 2.0 in 20k MWCO Slide-A-Lyzer dialysis cassettes (Cat# 66003, Thermo Scientific), concentrated on Amicon ultra filters (Cat# UFC810024, Millipore), and prepared for cryo-EM.

Size-exclusion purifications were carried out as follows. Lysates were prepared as described above and particles were pelleted and nutated in Phage Buffer 2.1 (same as Phage Buffer but with pH 7.4). The nutate was then spun at 12,000 × *g* for 10 min to remove aggregates then applied to an AKTA HiPrep 16/60 Sephacryl S-500 HR (Cat# 28935606, Cytiva) column and passed through at a flow rate of 0.5 mL/min. Protein content from elution peaks was assessed by SDS-PAGE/Coomassie, and samples containing PLE PLPs (elutions at 26–32 mL) were concentrated by centrifugation at 26,000 × *g* for 90 min, resuspended in ~100–500 µL Phage Buffer 2.0, and prepared for cryo-EM.

## Cryo-EM sample preparation and data acquisition

Cryo-EM samples were prepared by applying 5 µL aliquots of purified PLE PLPs to R2/2 Quantifoil grids and R2/2 Quantifoil grids coated with 2 nm ultrathin Carbon (QUANTIFOIL) that had been glow discharged for 45 s in a Pelco Easiglow glow discharging unit. The samples were plunge frozen in liquid ethane using a Vitrobot Mark IV operated at 4°C and 100% humidity, with a blot force of 1 and 5 s of blotting time per grid. The grids were screened at RTSF Cryo-EM facility using a Talos Arctica equipped with a Falcon 3EC direct electron detector and Cryo-EM data were collected at Purdue Cryo-EM facility using a Titan Krios equipped with a K3 direct electron detector, and operating at 300 keV with a post-column GIF (20 eV slit width) under low-dose conditions. Micrographs were collected at 64,000× nominal magnification (0.664 Å/pixel) by recording 40 frames over 3.1 s for a total dose of 36.41 e⁻/Å².

Data processing for icosahedral reconstruction of PLE procapsids was carried out using cryoSPARC v4.0.3. Briefly, the dose-fractionated movies were subjected to motion correction using patch motion correction with 2× binning, and CTF estimation of the resulting images was done using the patch CTF estimation jobs. The particles were picked using the Template picker job with templates generated from blob picker. Particles were then extracted and subjected to 2D classification. Cryo-EM image processing statistics, along with data collection parameters, are listed in *Figure 4—source data 1*. A total of 790,843 particles were used for 3D refinement with C1 symmetry, with a model generated from previous low-resolution data serving as the initial model. Refined particles were then subjected to a round of 3D classification in RELION 4.0 (*Kimanius et al., 2021*; *Asarnow et al., 2019*). A total of 379,643 particles were then imported into cryoSPARC (*Punjani et al., 2017*) and used for 3D refinement with icosahedral symmetry. The overall resolution was estimated using Postprocess job with a spherical mask in RELION 4.0 based on the gold-standard Fourier shell correlation (FSC) = 0.143 criterion (*Kimanius et al., 2021*; *delaRosa-Trevín et al., 2016*). The final map was sharpened with a B-factor of –130. The final maps were deposited into EMDB (accession number EMD-29675).

AlphaFold2 (*Jumper et al., 2021*) was used to generate homology models for all modeled protein chains. Initial models were then docked into EM maps for further refinement. For TcaP, only residues 34–172 were modeled. Refinement was carried out using Phenix (*Liebschner et al., 2019*) and model adjustments were carried out in *COOT* (*Emsley et al., 2010*). Model parameters were monitored using Molprobity in Phenix, and the values are listed in *Figure 4—source data 2*.

## Mass spectrometry

PLE and ICP1 PLPs subjected to mass spectrometry were further concentrated by trichloroacetic acid (TCA) precipitation, washed (10 mM HCl 90% acetone) three times, and air dried. The pellets were then resuspended in 100 mM Tris pH 8.5, 8 M urea, digested by trypsin, and analyzed by LC-MS/MS (Thermo LTQ XL linear ion trap mass spectrometer at the Vincent J. Coates Proteomics/Mass Spectrometry Laboratory at UC Berkeley). A sequence database containing *E. coli* proteins as well as ICP1 coat, scaffold, and decoration, as well as PLE TcaP, were used to compare the masses recorded and identify proteins in the sample.

## Acknowledgements

We thank the members of the Seed Lab for their useful feedback on this work. We would also like to thank the staff in the electron microscopy facility at the University of California Berkeley, RTSF Cryo-EM facility at Michigan State University, and the Purdue CryoEM facility for their assistance with the microscopy, and the staff in the Vincent J Coates Proteomics/Mass Spectrometry Laboratory for their assistance with sample processing. This work received funding from the J Coates Proteomics/Mass Spectrometry Laboratory at University of California Berkeley, the National Institutes of Health NRSA Trainee Fellowship (5T32 GM132022 to CMB in part), a National Science Foundation Graduate Research Fellowship (2018257700 to DTD), and a National Science Foundation. This work was supported by the National Institutes of Health (R01AI127652 to KDS; GM110185, GM140803, GM116789 to KNP); S10 Instrumentation Grant S10RR025622 in part for use of the Vincent CAREER Award (1750125 to KNP). Its contents are solely the responsibility of the authors and do not necessarily represent the official views of the National Institute of Allergy and Infectious Diseases or NIH. KDS holds an Investigators in the Pathogenesis of Infectious Disease Award from the Burroughs Wellcome Fund.

## Additional information

### Funding

| Funder | Grant reference number | Author |
| --- | --- | --- |
| National Institute of Allergy and Infectious Diseases | R01AI127652 | Kimberley D Seed |

| Funder | Grant reference number | Author |
| --- | --- | --- |
| National Institutes of Health | GM110185 | Kristin N Parent |
| National Institutes of Health | GM140803 | Kristin N Parent |
| National Institutes of Health | GM116789 | Kristin N Parent |
| National Institutes of Health | T32 GM132022 | Caroline M Boyd |
| National Science Foundation Graduate Research Fellowship Program | 2018257700 | Drew T Dunham |
| National Science Foundation | 1750125 | Kristin N Parent |

The funders had no role in study design, data collection and interpretation, or the decision to submit the work for publication.

## Author contributions

Caroline M Boyd, Conceptualization, Data curation, Formal analysis, Funding acquisition, Investigation, Visualization, Writing - original draft, Writing - review and editing; Sundharraman Subramanian, Data curation, Formal analysis, Investigation, Visualization, Writing - review and editing; Drew T Dunham, Funding acquisition, Investigation, Writing - review and editing; Kristin N Parent, Formal analysis, Funding acquisition, Writing - review and editing; Kimberley D Seed, Conceptualization, Supervision, Funding acquisition, Writing - original draft, Project administration, Writing - review and editing

## Author ORCIDs

Caroline M Boyd http://orcid.org/0000-0002-3966-9276
Kristin N Parent http://orcid.org/0000-0002-6095-0628
Kimberley D Seed http://orcid.org/0000-0002-0139-1600

Reviewer #1 (Public Review): https://doi.org/10.7554/eLife.87611.3.sa1
Reviewer #2 (Public Review): https://doi.org/10.7554/eLife.87611.3.sa2
Reviewer #3 (Public Review): https://doi.org/10.7554/eLife.87611.3.sa3

# Additional files

## Supplementary files

• Supplementary file 1. Escape mutation information. Phages genetically escaped TcaP's inhibition. These phages were purified and sequenced.

• Supplementary file 2. Mass spectrometry results of PLPs. ICP1 and PLE procapsid-like-particles were produced, purified, and subjected to mass spectrometry.

• Supplementary file 3. Oligos used in this study. Engineered plasmids and strains were constructed using these oligos.

• MDAR checklist

## Data availability

The data used for this work has been provided in the source data files associated with the figures. The cryo-EM data has been deposited to the EMDB under the following identification number: EMD-29675, and the PDB under the following identification number: 8G1R.

The following datasets were generated:

| Author(s) | Year | Dataset title | Dataset URL | Database and Identifier |
|---|---|---|---|---|
| Subramanian S, Boyd CM, Dunham DT, Seed KD, Parent KN | 2024 | A *Vibrio cholerae* viral satellite maximizes its spread and inhibits phage by remodeling hijacked phage coat proteins into small capsids | https://www.ebi.ac.uk/emdb/EMD-29675 | EMDB, EMD-29675 |
| Subramanian S, Boyd CM, Dunham DT, Seed KD, Parent KN | 2024 | A *Vibrio cholerae* viral satellite maximizes its spread and inhibits phage by remodeling hijacked phage coat proteins into small capsids | https://www.rcsb.org/structure/8G1R | RCSB Protein Data Bank, 8G1R |

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
