## [Editor Report · eLife assessment]

This **valuable** study reports on the structure and function of capsid size-determining external scaffolding protein encoded by a *Vibrio* phage satellite. The structural work is of high quality and the presented reconstructions are **compelling**. The paper offers a substantial advance in the field of phage and virus structure and assembly, with implications for understanding the evolution of phage satellites.

---

## [Referee Report · Reviewer #1 (Public Review)]

This paper describes the discovery, functional analysis and structure of TcaP, a protein encoded by the Vibrio phage satellite PLE, that forms a size-determining scaffold around PLE procapsids made from helper phage ICP1 structural proteins.

The system displays a fascinating similarity to the P2/P4 system, which had previously been unique in its use of a dominant, size-determining external scaffolding protein (Sid). An interesting observation is that PLE appears to be dependent on small capsids for efficient transduction, a phenomenon not previously seen in headful packaging phage/satellite pairs. It is not clear why this is the case.

The work is interesting, comprehensive and of high quality. The reconstruction and modeling statistics are good; unfortunately, although the map has clear alpha-helical density around the threefold axes, the TcaP model does not include this critical region. The comparison to Sid provides an illustration of probable convergent evolution.

The paper constitutes an important contribution to the field of phage and virus structure and assembly, with implications for understanding the evolution of phage satellites and for macromolecular assembly processes in general.

---

## [Referee Report · Reviewer #2 (Public Review)]

Phage satellites are fascinating elements that have evolved to hijack phages for induction, packaging, and transfer, promoting their widespread dissemination in nature. It is remarkable how different satellites use conserved strategies of parasitism, utilising unrelated proteins that perform similar roles in their cognate elements. In the current manuscript, Dr. Seed and coworkers elucidated the mechanism used by one family of satellites, the PLEs, to produce small capsids, a process that inhibits phage reproduction while increasing PLE transmission. The work is presented beautifully, and the results are astonishing. The authors identified the gene responsible for generating the small capsids, characterised its role in the PLE transfer and phage inhibition, and determined the structure of the PLE-sized small capsids. It is a truly impressive piece of work.

---

## [Referee Report · Reviewer #3 (Public Review)]

The manuscript by Boyd and co-authors "A *Vibrio cholerae* viral satellite maximizes its spread and inhibits phage by remodelling hijacked phage coat proteins into small capsids" reports important results related to self-defending mechanisms that bacteria are used against phages that infect them. It has been shown previously that bacteria produce phage-inducible chromosomal island-like elements (PLE) that encode proteins that are integrated into bacterial genome. These proteins are used by bacteria to amend the phage capsids and to create phage-like particles (satellites) that move between cells and transfer the genetic material of PLE to another bacteria. That study highlights the interactions between a PLE-encoded protein, TcaP, and capsid proteins of the phage ICP1.

The manuscript is well written, provides a lot of new information and the results are supported by biochemical analysis.